# Evaluation of Sanitation Strategies and Initiatives Implemented in Mexico from Community Capitals Point of View

**Thalía Turrén-Cruz [1],\*** , **Juan Alejandro García-Rodríguez [2]** and **Miguel Ángel López Zavala [1]**

1   Water Center for Latin America and the Caribbean, Tecnologico de Monterrey, Av. Eugenio Garza Sada Sur No. 2501, Col. Tecnologico, Monterrey, N.L., C.P. 64849, Mexico; miganloza@itesm.mx
2   Departamento de Ciencia Política y Relaciones Internacionales, Tecnologico de Monterrey, Av. Eugenio Garza Sada Sur No. 2501, Col. Tecnologico, Monterrey, N.L., C.P. 64849, Mexico; juanagr74@gmail.com
\*   Correspondence: thalia_turren@hotmail.com; Tel.: +521-961-17-25824

**Abstract:** Sanitation is fundamental to human development and well-being. For developing countries, such as Mexico, one of the greatest long-term challenges is to treat all the wastewater generated. Several projects have been implemented to achieve this goal, but, due to the idiosyncrasies of local people, they have failed. This study aims to analyze both, previous and current sanitation strategies and initiatives that are implemented in Mexico. Through the analysis of data reported by the literature, using the Community Capitals Framework (CCF) and focusing on human factors, the data was analyzed to identify whether communities are being considered to guarantee the success of the technologies and resources implemented. Besides the lack of information, it was understood that, despite the governance efforts to provide sanitation, the task remains incomplete; some top priority drivers, such as population growth and water supply, seem to define the progress or decline in providing quality basic sanitation services. Using the CCF, it was observed that financial, political, and built (infrastructural) are being prioritized over the human, social, cultural, and natural capitals. Therefore, it is important to highlight the communities' point of view on the development and decision making of projects and public policies, not just for sanitation but also for common well-being.

**Keywords:** Community Capitals Framework; public policies; sanitation services; sustainability; SDGs

## 1. Introduction

First, it is important to mention that the concept of sanitation here is focused to the collection and treatment of wastewater. In the period of the Millennium Development Goals (MDG's) 2000–2015 the global use of sanitation facilities rose from 54 to 68%. This falls below the 77% target, leaving 2.4 billion people without access to improved sanitation facilities; therefore, the challenge is huge due to the disparities between countries (Table 1) [1]. For this reason, The United Nations has defined a new initiative "Transforming Our World: The 2030 Agenda for Sustainable Development", defines the succession goals and targets of the MDG's, now called Sustainable Development Goals (SDG's). In this initiative, 2030 has been defined as the deadline to achieve access to adequate and equitable sanitation and hygiene services for all and to end the open defecation, paying special attention to the needs of women, girls, and people in situations of vulnerability [2]. Human development and well-being are defined among other factors by adequate nutrition, gender equality, education, and eradication of poverty; to achieve them, water and sanitation are fundamental [1].

**Table 1.** Access to drinking water and sanitation services of some countries [3].

| Country | Demography | | Access to Clean Water Services | | Access to Basic Sanitation Services | | |
|---|---|---|---|---|---|---|---|
| | Population (K) | % Urban | % with Access to Basic Water Service | Annual Rate of Change in Basic Provision | % with Access to Basic Sanitation Service | % of the Population with Open Defecation | Annual Rate of Change in Open Defecation |
| Argentina | 43,417 | 92 | 100 | 0.04 | 95 | 1 | 0 |
| Australia | 23,969 | 89 | 100 | 0.02 | 100 | 0 | 0 |
| Brazil | 207,848 | 86 | 97 | 0.25 | 86 | 2 | 0.86 |
| China | 1,376,049 | 56 | 96 | 1.22 | 75 | 2 | 0.96 |
| Germany | 80,689 | 75 | 100 | 0 | 99 | 0 | 0 |
| Honduras | 8075 | 55 | 92 | 0.66 | 80 | 7 | 1.19 |
| India | 1,311,051 | 33 | 88 | 0.48 | 44 | 40 | 1.5 |
| Japan | 126,573 | 93 | 99 | 0.03 | 100 | 0 | 0 |
| México | 127,017 | 79 | 98 | 0.6 | 89 | 2 | 0.86 |
| Nigeria | 182,202 | 48 | 67 | 1.42 | 33 | 26 | −0.26 |
| Russia | 143,457 | 74 | 96 | 0.07 | 89 | 0 | 0.3 |
| South Africa | 54,490 | 65 | 85 | 0.51 | 73 | 2 | 0.92 |
| Switzerland | 8299 | 74 | 100 | 0 | 100 | 0 | 0 |
| USA | 321,774 | 82 | 99 | / | 100 | 0 | 0 |

According to the National Water Commission of Mexico (CONAGUA), at the end of 2015 Mexico met the "sanitation" goal with 85% of its population covered with access to improved sanitation services; the national coverage of sewerage was 92.8% (97.4% urban, 77.5% rural); furthermore, 2447 wastewater treatment plants were in operation, treating 120.9 $m^3$/s of the 212 $m^3$/s collected [2]. This achievement has been a tremendous challenge because of the non-uniform distribution of water resources, central and northern parts of Mexico lacks of water availability (Tlaxcala, Aguascalientes, and Mexico City), while southeast has a water resource surplus (Chiapas, Veracruz, and Oaxaca) (Figure 1). It is important to underline that sanitation coverage reported by CONAGUA is referred to sewerage coverage and not exactly to sanitation (sewerage and treatment); therefore, if treatment coverage is considered, then the level of MDG's compliance will surely be significantly different. Though numbers about sanitation coverage are useful, there are still weaknesses such as poor quality service, bad operation and low collection efficiency, caused by politicization of services, lack of municipal autonomy, absence of coherent autonomy, and sector fragmentation, among others [4,5]. In order to improve sanitation, the construction of wastewater treatment plants in different parts of the country has been prioritized; which implies millions of dollars invested. These stunning engineering projects seem to be the solution, but, without taking into account traditional technologies and communities' point of view, it results in triggering social conflicts [6–8]. Accordingly, one of the great challenges in Mexico is to treat all of the wastewater generated. Despite tremendous efforts and achievements, several projects have not been successful due to the idiosyncrasy of local people and the mismatch between the value provided by the sanitation system and the values of people and/or communities; especially, of people in rural communities [9].

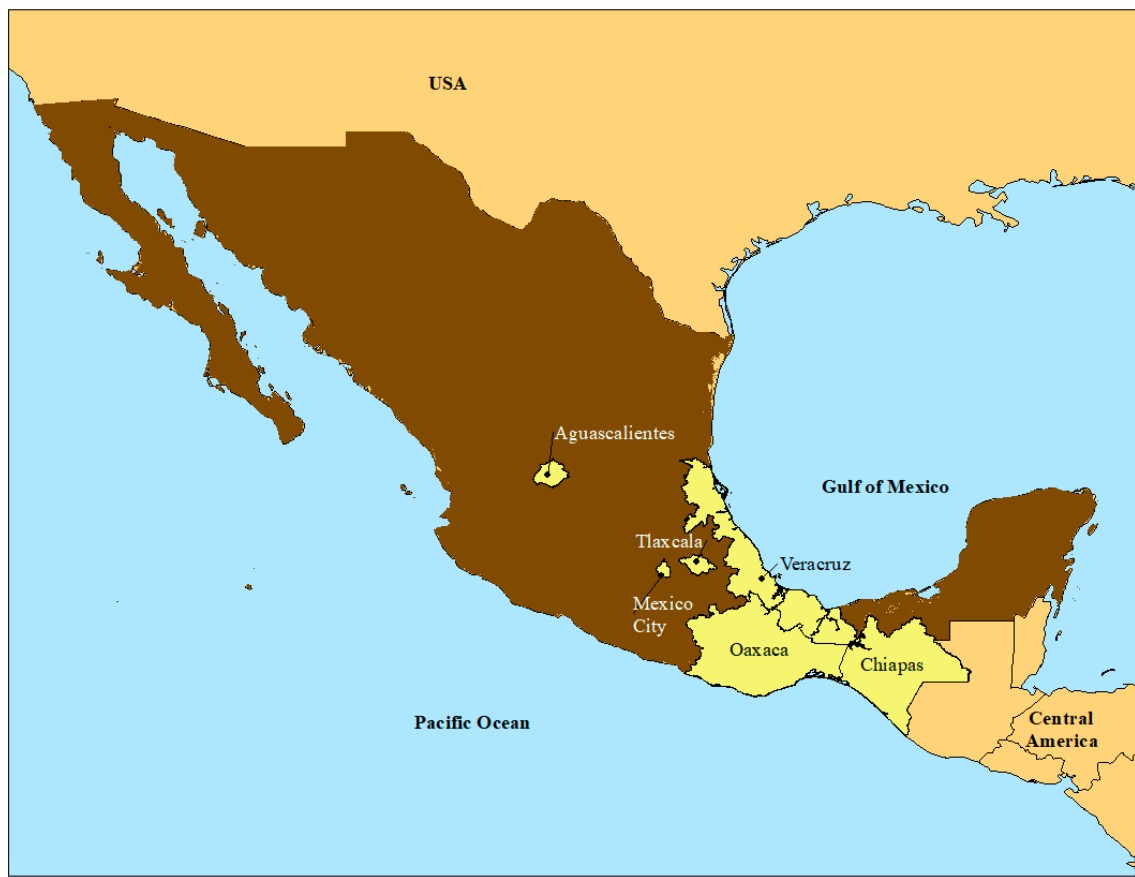

**Figure 1.** Contrasting water availability in Mexico, own elaboration with data from [10].

Consequently, in order to design, choose, and evaluate sanitation systems, sustainable criteria have been reported in the literature [11–15]. They divide the analysis into categories, such as environment, economy, technical function, socio-culture, and human, with a high emphasis on technical part. However, with the aim of having a complete and holistic comprehension of sanitation and all what it implies, the Community Capitals Framework (CCF) that was proposed by Flora et al. [16] is here used. CCF divides the analysis onto seven categories (capitals): Social, human, political, financial, built (from now on infrastructural) cultural, and natural. These capitals are defined as follows: Social capital involves mutual trust, reciprocity, collective identity, work together, and a sense of a shared future; it is interactive and an attribute of communities, not individuals. Human capital is referred to as the capabilities and potential of individuals, as determined by the intersection of genetics, social interactions, and the environment. It consists of the assets each person possesses: health, formal education, skills, knowledge, leadership, and potential. Political capital consists of organization, connections, voice, and power as citizens turn shared norms and values into standards that are codified into rules, regulations, and resource distributions that are enforced. Financial capital includes savings, income generation, fees, loans and credit, gifts and philanthropy, taxes, and tax exemptions. It is much more mobile than the other capitals and tends to be privileged because it is easy to measure. Infrastructural capital is human-constructed infrastructure. Although new infrastructural capital is often equated with community development, it is effective only when it contributes to other community capitals. Cultural capital determines a group's worldview, how it sees the world, how the seen is connected to the unseen, what is taken for granted, what is valued, and the potential possibilities for change. It includes the values and symbols reflected in clothing, music, machines, art, language, and ways of knowing and behaving. Natural capital includes the air, water, soil, wildlife, vegetation, landscape, and weather that surround us and provide both possibilities for and limits to community sustainability. It influences and is influenced by human activities. It forms the basis of all

other capitals. Capitals can contribute to or detract from communities from being sustainable, when a capital is highlighted over the others, resources are decapitalized, and the economy, environment, or social equity is compromised. CCF has been used in studies of various disciplines [17–22], giving the opportunity to comprehend how different areas and issues of interest are intimately related.

Here, CCF allows analyzing in the literature what has been and what is being done, focusing on human factors in order to identify whether people's needs are being considered to guarantee the success of the technologies and resources implemented on sanitation.

## 2. Materials and Methods

Recently, there have been numerous efforts on research about sanitation [11,13–15], but not much of them directly address the human side of sanitation in political, economic, environmental, cultural, infrastructural and social aspects.

A literature review was carried out following basically two steps. First, the inclusion and exclusion criteria were stablished to select the data to be reviewed. Second, the information obtained from literature was classified and analyzed according to CCF's concepts [16–22].

The Community Capitals Framework was revised and analyzed in order to understand the information to be collected. Then, search of secondary data sources, such as official government information, newspapers publications, and expert's opinions on sanitation was conducted. It was imperative to have several information sources, such as scientific and technical reports, and project's results. The search was conducted by accessing to different scientific websites, such as Scopus, Web of Science, Google Scholar, Research Gate, SciELO, Readlyc, and official governmental sites. In addition, references of selected data were explored to identify further relevant information. For the analysis, 89 documents were reviewed to identify and discuss the most important aspects of each capital's condition and their relations. Additionally, a synthesis with the most relevant descriptors was carried out.

## 3. Results

### 3.1. Human Capital

According to the most recent census, 43.6% of Mexico's inhabitants live in poverty and 22% are living in rural areas where there is a high index of unemployment, illiteracy, and most families live without access to water, sanitation, and energy services [23,24]. On Table 2, entities of the country with the highest and lowest levels of social delay and marginalization are shown, it can be appreciated that, states as Chiapas, Oaxaca, and Veracruz have the highest degrees of marginalization and social delay; and, the lowest percentage on sewage and piped water coverage.

As result, people are not usually aware of the effects of not having a sanitation system. For instance, some rural areas receive piped water, untreated or partially treated for use in daily activities and large-scale irrigation because they have no better options, deriving in malnutrition, anemia, and retarded growth [7,25] in places where the agricultural products are produced and sold. It reflects that a lag on hygiene and basic health conditions are becoming a national security issue as long as vulnerable people, such as infants and elders, are at risk [26,27]. The provision of drinking water and sanitation is a very important in population's health and they are also crucial for mortality and morbidity reduction in infants and waterborne diseases [28].

To prevent those problems, since 2001, Mexico has increased its investment in the development of drinking water, sewer, and sanitation infrastructure. Until 2015, sewerage coverage reached around 92.8%, and wastewater treatment plants treated around 60% of wastewater generated, with an average rate of change of 1.56% for drinking water and 3.28% for sewer coverage [28]. The increase in the coverage of these two basic services in the last 25 years has contributed to the reduction of mortality rate from 122.7 to 7.3 per every 100,000 inhabitants. The correlation between the reduction of mortality

with sewage coverage and water supply coverage is 0.98 and 0.99, respectively, which denotes that they are intimately related (Figure 2).

**Table 2.** Level of social delay and marginalization [2,28,29].

| State | Marginalization | | Social Delay | | Sanitation | Piped Water | Population in Mid 2016 |
|---|---|---|---|---|---|---|---|
| | Degree | Place in the Nation Context | Degree | Place in the Nation Context | % Households with Service | % Households with Service | (Millions of Inhabitants) |
| Chiapas | Very high | 2 | Very high | 3 | 87.2 | 88.6 | 5.32 |
| Oaxaca | Very high | 3 | Very high | 1 | 85.6 | 73.4 | 4.04 |
| Veracruz | High | 4 | Very high | 4 | 86.9 | 88.3 | 8.11 |
| Tlaxcala | Medium | 16 | Medium | 15 | 98.6 | 96.5 | 1.3 |
| Aguascalientes | Low | 28 | Very low | 29 | 99.1 | 98.8 | 1.3 |
| Mexico City | Very low | 32 | Very low | 31 | 98.6 | 98.8 | 8.83 |

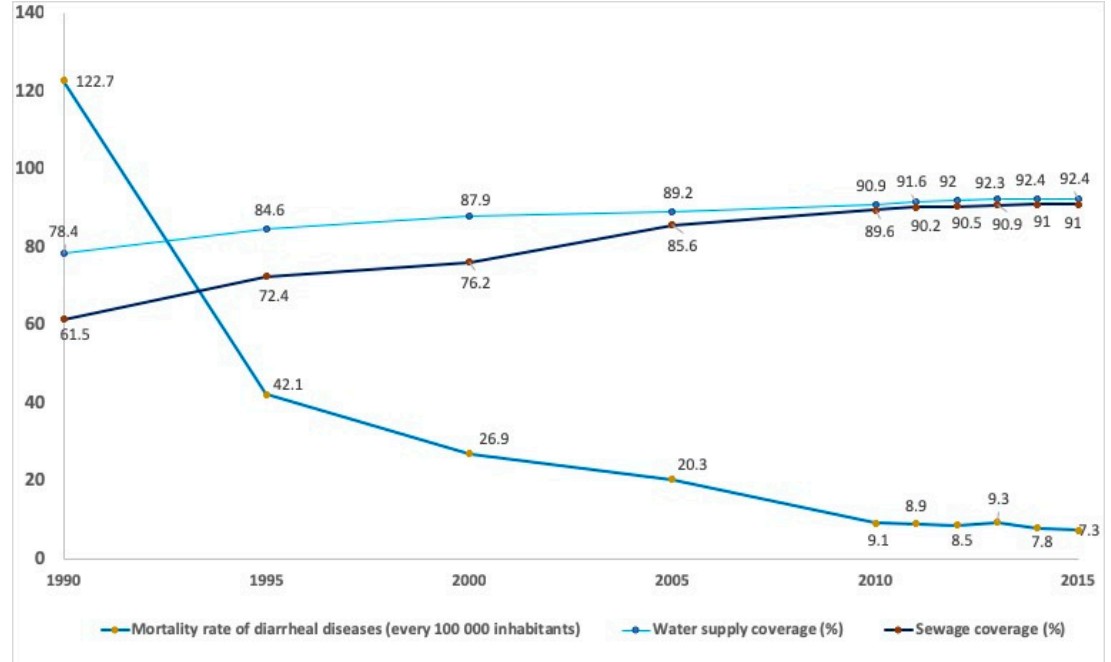

**Figure 2.** Mortality rate vs water supply and sewage coverage from 1990 to 2015 in Mexico; own elaboration with data from [28].

As it has been inferred in other countries [30], the construction of water and sanitation supply systems supports the control and eradication of epidemic outbreaks; but in Mexico, implementers seem to be incentivized to prioritize the construction of sanitation or water supply systems over ensuring a sustainable behavior change and ensuring the use of the systems. If communities continue growing with this lag, the possibilities for development are scarce [27,31].

On the other hand, the simple provision of water and sanitation does not guarantee better health. Access to sanitation is necessary to ensure water quality, but the provision of this service is more complicated, since it depends on many authority levels and it has a great impact on health and the environment [8]. It is important to mention that a sustainable sanitation system protects and promotes human health, is people-centered, responds to demand, does not contribute to environmental degradation or depletion of the resource base, is technically and institutionally appropriate, economically viable and socially acceptable, and it should have a high degree of functional robustness and flexibility [12,32]. Making people aware of the importance of sanitation is a beginning to encourage them to demand better services and to get them involved in the pursuit of better options.

*3.2. Social Capital*

Mexico is as big as diverse; it makes the distribution and availability of resources non-equitable, problems in communities are related to the necessity of satisfying basic needs, such as sanitation. Those communities have been forced to share their rivers, streams, and springs; furthermore, their lands have been polluted, flooded, eroded, and invaded [7]. Nevertheless, this is just to ensure the availability of a service.

In urban areas, it is almost impossible to ensure continuous access to water and sanitation for everyone due to demographic explosion that is caused by rural migration. As a result, rural communities are highly lagged and most of the time they are excluded from economical support because projects are not considering their low incomes [5,7,31,33]. It has been seen that the increasing pressure on resources triggers social conflicts related to their availability and exploitation. The provision of basic services seems to follow a path of taking from those who have the resources to give the ones who can pay for it, deriving in social lagging, lack of trust in public servers, also provoked by the limited efficiency on the sanitation services and policies [5,26,33,34].

In Mexico, the provision of sustainable water supply and sanitation systems to communities, mainly to those with high marginalization, is possible through social attention strategies that facilitate the appropriation of systems, the establishment of agreements, the consolidation of the organizational structure, and additionally, the link with the government to solve issues beyond local capacity [31].

However, the general social perception is that programs are top-down interventions with political interests [7,31]. In the attempt to tear apart mistrust, political interests, poor resource management, and to extend the cover of basic services and to improve communities' quality of life, in states like Veracruz and Chihuahua, self-management activities have been implemented [35,36]; also, non-governmental organizations, civil associations, and social organizations are working with communities on water management community systems through the implementation of good practices, social participation and training, implementation, and evaluation of projects, among others. Table 3 presents information related to some organizations, their objectives, and coverage level.

**Table 3.** Water organizations, their objectives and coverage level [37–39].

| Organization | Objective | Coverage Level |
| --- | --- | --- |
| National Water Coordinator for all, Water for life | Human right to water. | National level |
| FAN Mexico: Action Network for Water Mexico | Strengthening of local capacities. | National level |
| Water Ways (Caminos de Agua) | Create long-term sustainable water solutions. | Center/South |
| Coalition of Mexican Organizations for the Right to Water (COMDA) | Defense of water in favor of society and the environment. | National level |
| Blue Pitcher (Cántaro Azul) | Design, implement and evaluate integral services to guarantee sustainable access to safe water. | National level |
| National Association of Water and Sanitation Companies of Mexico (ANEAS ) | Support the elevation of efficiency in the provision of services. | National level |

It is important to mention that some of these organizations are integrated by many members, as is the case of COMDA, which encompasses 16 NGO'S and social organizations and an international ally (Latin America Office of the International Coalition for Habitat-HIC-AL) [37]. On the other hand, as can be appreciated, they all are focused on sustainable access to water, but not on sanitation. The projects of these different organizations have benefited the localities in Chiapas, Tabasco, Veracruz, Oaxaca, and Querétaro [38]. Others are focused on legal issues, such as the national water law, alternatives to privatization, the national consensus on water, and on the autonomy of indigenous peoples [39].

Rural communities are organizing to have active participation in projects that do not include them, striving for the use of alternative technologies, looking for sustainability, and trying to maintain and safeguard their resources [8,31,34]. Thus, involving the community throughout the entire process

seems to be a better option, as it could induce the sense of appropriation, and fees payment could be easily established if communities administer them [31]. Experts argue that the service has better quality in places where there is more society-government collaboration, if services are "brought closer to the people", then they fit to local needs and preferences [40,41].

### 3.3. Political Capital

Millions of people live in circumstances that represent considerable risk for their full development and healthy collective living. This is a consequence of the unequal distribution of public services between groups, since governments distribute them depending on their political goals and not for the social well-being [42]. In addition, sanitation generally receives less governmental priority when compared with the water supply. CONAGUA has delegated to municipalities, based on the 115 article of the Mexican Constitution, the authority to monitor and manage drinking water supply and sanitation, but since municipalities do not have governmental solidity to face this task, it has become an unmanageable problem [8,43]. Management instruments, fundraising, bilateral, and multilateral projects are focused specifically on water supply [28].

It can be said that Mexico has a robust legal framework, there are 22 Official Mexican Standards (NOM) related to water, but only five refers to sanitation as can be observed on Table 4.

This unfair distribution of political instruments is a huge challenge due to the lack of political willingness to face sanitation challenges. The fact that government interventions are mostly top-down and the lack of consensus between sectors, political parties, government, and society, results in the benefit of only few people. All this together with corruption, clientelism, paternalism, deficiency of legal-coherent policies (national and municipal), lack of municipal autonomy and technical capacities, and lack of not long-term projects, discourages community participation and develops a conflicting framework and sector fragmentation [4,5,34,44,45]. It is important to recognize the right to clean water and sanitation, which implies establishing solid governmental mechanisms to guarantee communities' well-being.

**Table 4.** Official Mexican Standards [46–50].

| Ministry of Environmental and Natural Resources (SEMARNAT) | National Water Commission (CONAGUA) |
|---|---|
| • NOM-001-SEMARNAT-1996<br><br>Maximum permissible limits of contaminants in wastewater discharges in national waters and goods. | • NOM-001-CONAGUA-2011<br>Drinking water systems, domiciliary sewer and sanitary sewer-Hermeticity-Specifications and test methods |
| • NOM-002-SEMARNAT-1996<br>Maximum permissible limits of contaminants in wastewater discharges to urban or municipal sewerage systems. | **Ministry of Health (SSA)**<br><br>• NOM-230-SSA1-2002 |
| • NOM-003-SEMARNAT-1997<br><br>Maximum permissible contaminant limits for treated wastewater that is reused in public services. | Sanitary requirements for water management in drinking water networks. |

The previous, together with encroachment that is associated with population growth and migration, and irregular settlements makes the situation even more complicated. Unfortunately, there is no government willingness to solve this situation, excused in "legal issues" [26,51,52]. Also, we must add that administration period (national, state, and municipal) is short when compared with the time that is needed to implement and monitor projects. Furthermore, short and long-term tracking mechanisms are not being considered to ensure the continuity of projects [31,35], without forgetting to urge solid self-management laws and activities, to encourage communities' participation. Therefore, projects are probably achieving their technical purpose but not the human: covering a basic need of population to improve their quality of life. Mexican institutions have placed additional effort on

allowing private investment, because municipalities are not capable of maintaining their drinking water and sanitation systems. To allow private participation, reforms to policies have been made leading to investment of dubious origin and for the benefit of a few [7,45,53]. It seems that changing or improving regulations and policies is motivated merely by political interests. Sanitation is not a priority, there is no coordination between government levels, participatory and stakeholder models are not efficient, norms are not applied, there is no transparency in the application of policies, and different political ideologies create barriers instead of tearing them apart. This shows the absence of a real interest on improving the sanitation systems [8,54]. It is important to highlight the efforts, but it is also important to say that it is not been enough; municipalities have not achieved their task to provide basic services according to that established in the Mexican constitution.

### 3.4. Financial Capital

Providing basic services to communities, particularly to rural communities, has been complicated, especially when considering Mexico's economic crises. Sometimes solutions to a certain issue equal the investment to extend the sanitation system coverage, meaning a strong financing to a single locality; deriving in expanding the system instead of improving and making it more efficient [31]. In financial terms, it costs less to build new infrastructure.

As municipalities do not have capacities to provide services of quality, no real fees are settled, the user is not willing to pay because money is not well managed and invested, and low budgets and minor investments affect the quality of the service and infrastructure [4,55]. In order to support municipal responsibilities, the Federal Government through programs, such as "Drinking water, sewerage and sanitation in urban zones" (APAZU for its initials in Spanish) and "Drinking water and sanitation in rural communities" (PROSSAPYS for its initials in Spanish) aims to face needs in both urban and rural areas through the economic support of the Inter-American Development Bank (IDB); observe on Table 5 that investment also covers water supply, using less than 26% of the total investment to improve sanitation systems. It is comprehensive since access to water has been considered as an urgent matter and sanitation has been put in a second place.

**Table 5.** Investment on drinking water and sanitation in rural communities (PROSSAPYS) [56].

| Project Stage | Number of Projects (Thousand) | | Number of Persons Benefited | | Investment (USD) | Period |
|---|---|---|---|---|---|---|
| | Potable Water | Sanitation Systems | Water | Sanitation | | |
| PROSSAPYS I | 3, 950 | 1, 325 | 2.16 million | 643 thousand | | 2001–2005 |
| PROSSAPYS II | 1, 933 | 578 | 745 thousand | 430 thousand | $ 292, 500, 000 | 2005–2007 |
| PROSSAPYS III | 1, 940 | | 1.5 million | | $ 100, 000, 000 | 2011–2013 |
| PROSSAPYS IV | | | | | $ 571, 000, 000 | 2014–2017 |

Here, it is important to underline that investment is focused on water treatment in urban areas and in rural areas on drinking water and sewerage [52]. Accordingly, it is important to realize that priorities are different between regions, and every project must be designed and applied according to the needs and local requirements. The above confirms there are efforts to extend the provision of basic services, but there are still important issues to improve to make projects succeed.

The government through private investments works in large-scale projects that require high technology instead of sustainable low-cost technologies [44], making them difficult and expensive to maintain and since laws, regulations, and projects are not solid, there is a continuation of unsuccessful projects. There is no investment on monitoring, the lack of the service turns into economic problems, if service is not "good" people do not pay and there is no money to keep giving the service, so it becomes a never-ending cycle [4,5,45]. It allows for the authorities to justify the "need" of private

investment, but as there are no companies with sufficient economic solvency in the country, foreign companies in association with local capitalists are "solving" the country's problems [45]. People with lower incomes who struggle to obtain clean water are worried about the economic impact of those investments and think they are going to have to pay more for basic services [7,27], and they probably will. In places where self-management activities have been developed, their sustainability has not been possible due to insufficient government support and the lack of human and financial resources [35]. As can be appreciated, this capital is intimately related with the political, infrastructural, human, and social capitals.

### 3.5. Infrastructural Capital

Municipalities have tried to fulfill its function of providing the service in a partial manner, despite the large number of localities and operative-financial restrictions [27,31,45]. In this matter, the most common facilities implemented through the Infrastructure National Program 2014–2018 (with private participation) are wastewater treatment plants in the Valley of Mexico, Sonora, Chiapas, Hidalgo, San Luis Potosí, Chihuahua, Nayarit, Guerrero, Veracruz, Quintana Roo, and others [2,6].

It implies that millions of Mexican pesos invested and yet people do not have access to services with quality [7,8]. Apparently, the great engineering projects are the solution in urban areas. Despite an increase in number, there is a lack of proper operation related to their excessive mechanization, instrumentation, automation, and a high demand for electricity, which increases their maintenance and operation costs. In addition, their technical personnel are not always adequately trained and their payments are low. Moreover, some plants are not appropriate to the climate and physical conditions of the place where they are established; causing a low amount of wastewater that is treated without correct final disposal [44]. By the end of 2015, 2,447 wastewater treatment plants were in operation, treating 120.9 $m^3$/s of the 212 $m^3$/s collected through sewage systems, this means that around 40% of wastewater without treatment is being disposed into land or water bodies [23]. Compared with numbers in 2010, 87.31 $m^3$/s treated of 208 $m^3$/s collected through sewage system [52]; the evolution in five years is quite good but still not enough. Other technologies implemented (mostly in rural areas) with less investment and popularity are latrines, septic tanks, bio-digesters, irrigation systems, and wetlands [7,31,44,52]. These technologies seem to be less innovative and applicable just for rural and poor areas, and it is said these are the most recommendable systems in most cases. Nevertheless, no systems are ensured success if they are designed without considering traditional technologies and community's point of view, the specific conditions of the area of implementation (local context), and constant training of the human resources to keep technologies functioning [5,12,52,57]. In places such as Cozumel and Playa del Carmen, there are some urban areas that were developed in the 90's that still have no full cover of drinking water and sanitation systems, due to the pressure of migration and the creation of new urbanized centers [58]. This is supported by [34], who says, "not all large populations due to the explosive population growth are able to maintain self-sufficient sanitation systems". Current legislation and decision-making procedures for choosing sanitation systems seem to be based mostly on initial investment and the operation and maintenance costs of sanitation systems.

### 3.6. Cultural Capital

The achievements on access to drinking water and sanitation by local and national governments have been possible, not just because of governmental efforts, but because people have been able to adapt or accept what has been imposed. However, it must not be forgotten the failure of the projects also derives from the acceptance of the people who usually does not have the cultural understanding that sanitation is very important [59], as discussed in human capital. The debate of providing sanitation could be considered to be a cultural issue about resistance, consent, and education. Thus, to provide an adequate service to a community, many things need to be understood: community's point of view, knowledge of the ecosystem, the importance of the service to the community, its agricultural activities, and the water management culture evolution over the history between others [60,61]. In this

regard, it is important to clarify that a water culture paradigm can be described as the "continuous process of production and updating of individual and collective transformation of values, beliefs, perceptions, knowledge, traditions, aptitudes, attitudes and behaviors in relation to water in daily life" [59,62,63]. Having defined this, some of the population sectors are usually found to have higher disadvantage when facing water and sanitation problems, including women, urban marginalized communities, and particularly indigenous' and rural communities [64–67]. Indigenous communities that are more commonly affected by infrastructural projects, appropriation, and over exploitation of their natural resources have formed different movements as United Towns for the Care and Defense of Water, Front of Indigenous Peoples in Defense of the Mother Earth, Front 9 June in Defense of the Natural Resources, Fire of Dignified Resistance, Zapatista Army of Women in Defense of Water, just to mention some. They are fighting for their right to manage and protect their own resources, through legal defense, community strengthening, promoting the leadership and participation of young people, women and indigenous communities, and influencing governments to guarantee the rights of all [68,69]. Most of them have succeed in the fight but have also been attacked through governmental instruments or institutions that attempt to force them to give up their fight, as the case of indigenous municipalities in the State of Mexico, like Coyotepec, Tecámac and San Salvador Atenco, Valley of the Mezquital; La Parota in Guerrero, Montes Azules in Chiapas, and many others. Through the desiccation of lagoons and wetlands, the modification of rivers, the flooding of villages and agricultural areas by the construction of hydroelectric plants, the contamination of basins, rivers, and aquifers by the oil industry, the dispossession of springs, and the transfer of water to large cities are just some of the examples of government decisions that have affected indigenous peoples and peasant communities in various states and regions of Mexico [70]. These communities must defend the resources that they need to sustain their livelihoods, in the face of governmental decisions that do not consider their needs and rights.

The inclusion of indigenous communities has been difficult due to radicalization in the attachment to their customs (indigenous people is not meant to participate in politics in many cases and women are not allowed to participate at all), there is difficulty in communication due to diverse languages, mistrust to the authorities, dispersion, extreme poverty, and difficult access to communities [31]. Some other communities care little about those services or they are very ingrained in the belief that they will not pay for the services that the government provides. In addition, heterogeneous communities have more problems in the provision of services than homogenous areas. Nevertheless, "social punishment" to government using or not using some services is one of the crucial factors to the access to sanitation added to the water's use and management mentioned above [31]. Since the structural and financial capacity of municipalities is low, many communities use pit latrines and open defecation. Eradicating open defecation is recognized by the World Health Organization as a top priority for improving health, nutrition, and productivity of populations [1]. The constitution of Mexico, in its 4th article, establishes the need to adopt special measures to safeguard people, institutions, goods, work, cultures, and the environment of indigenous people [43,71]. However, these sanitation services are defined as a second human priority [59].

Furthermore, improvement of communities' sanitation system is limited by land tenure and its legal status is not clear, as said government is not supporting them, resulting in the indigenous and marginalized communities most affected, not just by land irregularities also because of their difficulty to communicate in their own languages. Most of the time, they are represented by a leader who receives privileges and ultimately seeks for his/her own interests. Ultimately, people are not paying for the service because it is not fulfilling their needs [7,45].

Some organizations have identified that the prevailing western vision does not understand or simply ignores the cultural and spiritual vision of indigenous people. These communities are not significantly included in the policy processes and planning, their livelihoods are frequently ignored by authorities, and the bodies of water and land they use to subsist are being polluted by external forces [67]. Finally, the cultural differences surrounding access to sanitation in Mexico have been

caused by social inequality, since this issue is related to many factors, such as ethnicity, class, culture, lifestyle, and political power in decision-making [71].

*3.7. Natural Capital*

First, there is a huge contrast in terms of the availability of resources and coverage of services in the country due to geographic context, precipitation's distribution, type and amount of drinking water sources, droughts, and many other factors [28]. Central and northern parts of Mexico lacks of water availability (Tlaxcala, Aguascalientes and Mexico City), while southeast has a water resource surplus (Chiapas, Veracruz and Oaxaca); 61% of the water available is condensed in five of the 32 states of Mexico (Table 6). In Table 7 the drinking water consumption per capita is appreciated along with the amount of wastewater treated. The contrast between drinking water per capita and water sources in the case of Mexico City can be explained by the water that is imported from hydrological basins nearby [7].

Subsequently, with the urgency to satisfy basic needs, such as sanitation, coverage of sewage services has been prioritized over installation of treatment systems. This means about 40% of domestic wastewater that is collected is discharged without treatment into rivers and streams, or they are directly used for agriculture, negatively affecting health and the ecosystems [52,72].

**Table 6.** Contrasting water sources in Mexico [73–75].

| | Number of Aquifers | Groundwater Availability (hm$^3$/year) | Status | Number of Wells | Surface Water Bodies | Number of Rivers |
|---|---|---|---|---|---|---|
| **Total National** | 195 | 22.509 | Deficit | 14,326 | 1099 | 2234 |
| | 458 | | Availability | | | |
| **State** | | | | | | |
| **Chiapas** | 15 | 2.993 | Availability | 653 | 75 | 167 |
| **Oaxaca** | 21 | 429 | Availability | 507 | 24 | 165 |
| **Veracruz** | 18 | 1.175 | Availability | 807 | 101 | 117 |
| **Tlaxcala** | 4 | 101 | Availability | 90 | 1 | 4 |
| **Aguascalientes** | 5 | * | Deficit | 93 | 6 | 7 |
| **Mexico City** | 1 | * | Deficit | 52 | 1 | 4 |

Note: * non-available information.

**Table 7.** Contrasting drinking water use and domestic wastewater treatment in Mexico [2,28].

| State | Drinking Water 2016 | | Treatment of Domestic Wastewater | | Population in Mid-2016 |
|---|---|---|---|---|---|
| | Total Consumption (hm$^3$/year) | Per Capita (m$^3$/hab/year) | Installed Capacity (hm$^3$/year) | Treated Discharge (hm$^3$/year) | (Millions of Inhabitants) |
| Chiapas | 389.0 | 73.12 | 60.64 | 40.52 | 5.32 |
| Oaxaca | 266.3 | 65.91 | 50.39 | 33.77 | 4.04 |
| Veracruz | 551.3 | 67.97 | 232.95 | 164.55 | 8.11 |
| Tlaxcala | 89.5 | 68.84 | 46.23 | 30.21 | 1.30 |
| Aguascalientes | 127.1 | 97.7 | 154.84 | 52.47 | 1.30 |
| Mexico City | 1089.6 | 123.39 | 242.48 | 176.7 | 8.83 |

With the aim to solve some problems in communities, the wastewater plants constructed represent a larger problem than solution, with the focus on conventional methods, such as activated sludge, which requires the intensive use of chemical products and energy, generates emissions of contaminants, such as ammonia and significant amounts of sludge that do not have adequate safe final disposal locations [72]. Additionally, wastewater is transported to the field or river far from

cities or communities (in the best cases) and it causes the contamination and deterioration of surface and groundwater quality, as well the resources as soil and food. This causes loss of resources and availability competition [5,7,34,55]. It is evident that infrastructure and financial capacity is exceeded by the needs of the country. Institutional incapacity to manage resources has caused rivers' pollution, and a deficit in the provision of services and lack of wastewater treatment [35]. An example of this is that six out of 10 rivers in Mexico are contaminated with some type of pollutant [76]. These bodies of water are a crucial resource not just for the environment, but also for communities that live in the same water basin, such as for indigenous people who are one of the closest social groups to rivers [65]. To 2016, according to [77], wastewater discharges were not inspected in five of every six aquifers, which implies that water authorities are not aware of whether contaminants that compromise water quality were verted in more than half of country's aquifers, risking human health and productive activities. Around 70% of surface water bodies in Mexico have a degree of contamination, Table 8 shows points of discharge of municipal wastewater without treatment according to type of receptor body at the national level.

**Table 8.** Municipal Wastewater discharge points [77].

| Total Points | Amount of Discharge Points by Type of Body Receptor * | | | | | | | | |
|---|---|---|---|---|---|---|---|---|---|
| | Sea | Lake or Lagoon | River or Stream | Dam | Channel or Drain | Ground or Canyon | Collector | Other Body Receptor | Not Specified |
| 4887 | 8 | 249 | 2461 | 40 | 594 | 972 | 150 | 401 | 12 |

\* Receiving municipal untreated wastewater.

On the other hand, natural resources can become a threat to human populations and they can be shaped by the impact humans make on ecosystems and man-made systems. Natural hazards are not only caused by natural phenomena, but also the outcome of socio-environmental changes over time, such as urbanization, migration, and capitalist development [78]. Federal water management authorities have repeatedly expressed that the treatment of wastewater in Mexico should be one of the main strategies for preserving water quality. However, improving the quality of life, protecting public health, and guaranteeing sustainable development have not been improved, despite the increase in investment. The country cannot continue affording wrong management of its natural resources [76].

## 4. Discussion

It is important to highlight the lack of information about sanitation in Mexico. Official reports from different governmental programs (PROSSAPYS, APAZU, etc.) are focused on drinking water and sewage, with sanitation being merely mentioned rather than prioritized. On the other hand, most scientific information is focused on the same way [23,79,80], but partially analyzing self-management activities, social perception, and willingness to pay services, mostly in urban areas. There are some exceptions, like [52], comparing water treatment in both rural and urban areas; and [58], with the analysis to access to drinking water and sanitation.

It is also important to mention that sanitation problems are not exclusive of Mexico. In Latin America, Africa, Asia, and Europe experience, limited access to drinking water and sanitation increases the incidence of gastrointestinal diseases of infectious origin, which in marginal and indigenous localities can lead to death [22,30,58,81–84]. In the case of Colombia, the government has invested 1100 US million dollars in the start-up of wastewater treatment systems from 2011 to the first half of 2013, however, incidences of water-borne diseases, such as acute diarrheal, food-borne diseases, and fever typhoid and paratyphoid, have not decreased in the period 2008 to 2014. Only hepatitis A has decreased [82]. In Luxemburg, sanitation issues are faced with common solutions, such as building large sewer systems and new centralized treatment plants and continually upgrading and expanding existing infrastructure [83]. These methods not only imply important costs and investments,

but also have significant environmental impacts, as the small receiving water bodies are vulnerable to discharges from wastewater treatment facilities.

As an alternative to conventional systems, ecological sanitation (Ecosan) has been introduced in China, Uganda, and Mozambique; urine diverting toilets in India and Sri Lanka; dry toilets in Japan and dry composting toilets in hot arid climates, just to mention a few. Most of them have been successful with comprehensive and holistic approaches cooperating with different areas interested on diverse issues [84]. According to Benetto et al. [83], Ecosan seems to have significant advantage as compared to conventional wastewater treatment systems regarding the reduced contribution to ecosystem quality damage. However, it seems to generate higher impact on climate change and human health than conventional treatment systems due to the emission of some gases during the transport and process of urine and compost (ammonia and nitrogen) that contribute to acidification and global warming; and, the production and end life management of some devices (bio-filter, separation toilets); all of them must be improved to decrease any negative impact.

Meanwhile, ecological sanitation is a promising alternative to small-scale wastewater treatment. At this scale, nutrients flow and losses can be better managed, secondary fertilizer can be easily used by interested farmers and have potential economic value, with related advantages on the environment [83]. There are many other options, such as anaerobic co-digestion of excreta and organic solid waste that may be feasible to produce biogas or grey water management in urban slums [55]. Nevertheless, it is important to emphasize that all sanitation technologies must be tested at the pilot scale to evaluate their performance prior to scaling up for wide application. Equally important, the acceptability by the users and existence of an enabling institutional framework improve chances of technology success, since sustainability requires that institutional arrangements and collection of the waste, treatment, reuse, and safe disposal complement each other [55].

Following the line where acceptability turns into an important factor of success becomes important to mention the case of some implemented projects. For example, Sinha et al. [30] assessed the patterns and determinants of latrine use in India, where a governmental program was executed to accelerate sanitation coverage in rural areas and end open defecation. In the period from 1999 to 2010, 64.3 million latrines were constructed. They found that not all individuals living in households with access to latrines use the system, which suggests that the coverage does not necessarily translate into use. This implies that focus should also be placed on the use of the technology rather than only on access and coverage. This confirms Wicken's analysis [85], which says that the access to improved sanitation infrastructure is being used merely as an indicator, resulting in the construction of millions of latrines that may or may not lead to improved sanitary outcomes and health benefits, since the use of them is not being assured. Thus, sanitation means more than having access to a latrine; it includes excreta disposal, solid waste, and wastewater management and hygienic practices. Millions of latrines are being constructed each year and yet the frequency of their usage is unknown. Infrastructure is being implemented without sustainable behavior change. This exhaustive work in increase sanitation coverage is additionally related to the change of MDGs to SDGs, where there are 17 new targets to achieve. Focusing on target six: "Ensure availability and sustainable management of water and sanitation for all" through access to safe and affordable drinking water, adequate and equitable sanitation and hygiene, halving the proportion of untreated wastewater, among others [86]. However, if we evaluate how target six is related to other targets, it can be seen it is directly associated to goals one, three, and eleven, their relation could hinder the achievement not of a single goal, but all of them. The goals are focused on ending poverty in all its forms by ensuring equal access to basic services, reducing the number of deaths and illnesses from air, water, and soil pollution and making cities inclusive, safe, resilient, and sustainable. This interrelation between SDGs also helps to understand the interrelation between the capitals in the CCF and how a weak capital in a community affects all of its capitals, ultimately avoiding them for reaching sustainable development.

According to [2,22], communities are poor when the stocks of various capitals are low. Moreover, to face poverty, the investment on natural, cultural, and human capitals must lead the formation of

social, political, financial, and infrastructure capital. With the analysis that was conducted in this paper, capitals such as the financial, political, and infrastructure are being prioritized over the natural, human, social, and cultural. This compromises the availability of resources and a quality of life, leading communities to poverty. To reduce the communities' poverty, it is necessary to identify and transform the community's capitals for sustainable development; at the same time, it is also necessary to address poverty as a community issue, and to seek place-based, rather than individual solutions, with the intention to understand and engage all of the capitals to set the basis of poverty reduction [22]. Based on the above, social participation at local levels can help communities to comprehend and value their participation on solving their problems, using tools such as focus groups, workshops, ethnographic studies, and surveys, among others; with local people and stakeholders to make them aware about their responsibility and participation in the design of solutions [17,87–89], as a form of empowerment and involvement in improving their quality of life. In addition, human behavior, adaptation, customs, habits, and livelihoods of communities must be added and emphasized to the design and implementation of sanitation systems to ensure that they are going to be holistic, sustainable, and successful.

Finally, Figure 3 shows the current pyramidal structure of how sanitation works in Mexico. To achieve a holistic structural change, it must tear apart the current structure and focus on creating balance in all capitals.

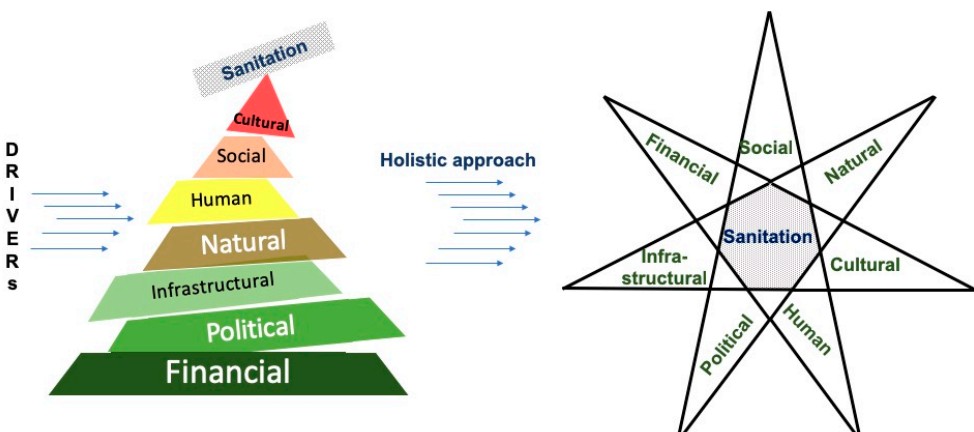

**Figure 3.** Current sanitation's structure vs Proposed holistic approach analysis (own elaboration).

## 5. Conclusions

Evaluating sanitation through CCF allows for identifying the disparities between capitals, there is no social cohesion, government policies, and strategies are not reliable. Health is being affected and weakening human capital, corruption, and all its forms are deeply rooted in political capital. There is no appropriation of the strategies implemented causing wasted economic resources and the pollution of the environment. Financial, political, and infrastructure capitals are being prioritized over the natural, human, social, and cultural, which are compromising the availability of resources and a good quality of live. Drivers that are related to the lack of improved sanitation services in Mexico are population growth, drinking water supply as a top priority, political factors, poor planning, and poor financial resources' administration.

These findings indicate a direct and complex relationship between the capitals and the provision of basic sanitation services along the country, associated to the inequitable distribution and availability of several resources, mainly in states where the conditions of poverty have been maintained for a long time in the absence of an adequate political framework, the misuse of their resources, among other things, which have caused the slow human, social, and economic development. This in contrast to states with better development conditions but scarce resources availability.

The design and application of holistic studies and frameworks is imperative to estimate the real situation about sanitation in Mexico and potentially worldwide. It is urgent to change the actual way to solve problems, analyzing them as a whole to solve them in a holistic way.

**Author Contributions:** Investigation, T.T.-C. and J.A.G.-R.; Writing – original draft, T.T.-C. and J.A.G.-R.; Review&Editing, T.T.-C. and M.Á.L.Z.; Funding Acquisition, M.Á.L.Z.

**Funding:** This research was supported by the National Council of Science and Technology of Mexico (CONACYT) and the Tecnologico de Monterrey.

**Conflicts of Interest:** The authors declare no conflict of interest. The funders had no role in the design of the study; in the collection, analyses, or interpretation of data; in the writing of the manuscript, or in the decision to publish the results.

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
