# Peer review of "Evaluation of Sanitation Strategies and Initiatives Implemented in Mexico from Community Capitals Point of View"

_water, doi:10.3390/w11020295_

Round 1

Reviewer 1 Report

The manuscript water-430516 concerns the evaluation of sanitation strategies implemented in Mexico with the Community Capitals approach.

I think that the results are interesting and can be useful for all the scientific community involved in the "water, sanitation and hygiene" - WASH topic.

I read carefully the manuscript and I think it can be accepted with minor revisions.

Comments and suggestion:

Section Introduction: only for comparison add also a new table reporting data of sanitation and drinking water of several world countries.

Section Introduction: add the political map of Mexico in which the States of table 5 are reported.

Section 2: add references.

Table 1 and 5: change “renewable water” with “drinking water”.

Table 5: add also the sanitation water for the 5 states in hm3/year.

Section 3.7: add a table reporting for all the Mexico Nation and the representative states of table 5 the drinking water sources in hm3/year such as lakes, artificial reservoirs, wells, river, springs, etc. Then, describe the new table in the text.

Reword the conclusions adopting the key-point approach.

Author Response

Section Introduction: only for comparison add also a new table reporting data of sanitation and drinking water of several world countries.

Answer: A table has been added.

Section Introduction: add the political map of Mexico in which the States of table 5 are reported.

Answer: A map and description have been added.

Section 2: add references.

Answer: The references have been added.

Table 1 and 5: change “renewable water” with “drinking water”.

Answer:

In table 1 data about sanitation and piped water have been added instead “Renewable water” and in table 5 data about Renewable water has been changed by data of drinking water coverage, according with CONAGUA.

Table 5: add also the sanitation water for the 5 states in hm3/year.

Answer: The data has been added to the table.

Section 3.7: add a table reporting for all the Mexico Nation and the representative states of table 5 the drinking water sources in hm3/year such as lakes, artificial reservoirs, wells, river, springs, etc. Then, describe the new table in the text.

Answer: Table 7 has been added in the reviewed manuscript with the available data on water sources.

Reword the conclusions adopting the key-point approach.

Answer: This section has been improved.

Thank you for your comments

Reviewer 2 Report

Based on the figure 1 You can calculate:

1. Rate of change or trend equation for Mortality rate, Water supply coverage (%) and Sewage coverage (%)

2. correlation coefficient between:

Mortality rate and Water supply coverage

Mortality rate and Sewage coverage

Author Response

Based on the figure 1 You can calculate:

1. Rate of change or trend equation for Mortality rate, Water supply coverage (%) and Sewage coverage (%)

2. correlation coefficient between:

Mortality rate and Water supply coverage

Mortality rate and Sewage coverage

Answer:

Calculations have been added in the revised manuscript.

Thank you for your comments
